# Constant Transmission Efficiency Dimming Control Scheme for VLC Systems

**Jia-Ning Guo, Jian Zhang \*, Gang Xin and Lin Li**

National Digital Switching System Engineering & Technological Research Center, Zhengzhou 450000, China;
14291003@bjtu.edu.cn (J.-N.G.); downxg@163.com (G.X.); wclilin@163.com (L.L.)
\* Correspondence: Zhang_xinda@126.com

**Abstract:** As a novel mode of indoor wireless communication, visible light communication (VLC) should consider the illumination functions besides the primary communication function. Dimming control is one of the most crucial illumination functions for VLC systems. However, the transmission efficiency of most proposed dimming control schemes changes as the dimming factor changes. A block coding-based dimming control scheme has been proposed for constant transmission efficiency VLC systems, but there is still room for improvement in dimming range and error performance. In this paper, we propose a dimming control scheme based on extensional constant weight codeword sets to achieve constant transmission efficiency. Meanwhile, we also provide a low implementation complexity decoding algorithm for the scheme. Finally, comparisons show that the proposed scheme can provide a wider dimming range and better error performance.

**Keywords:** visible light communication (VLC); dimming control; constant transmission efficiency; error performance; light-emitting diode (LED)

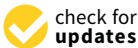



## 1. Introduction

With the increasing shortage of radio frequency spectrum and the development of light-emitting diodes (LEDs), visible light communication (VLC) has attracted extensive attention from many scholars [1,2]. Compared to conventional wireless communications, VLC has higher rates, lower power consumption, and less electromagnetic interference [3–5]. As a combination of communication and illumination, VLC achieves reliable communication and high-quality illumination through the fast response characteristic of LEDs. For convenience, VLC systems realize high-speed communication utilizing intensity modulation and direct detection (IM/DD). There is no doubt that VLC is a novel wireless communication mode with great potential. At the same time, illumination function plays an essential role in the indoor VLC system. Dimming control is one of the most crucial illumination functions that can adjust the brightness according to the users' requirements under the condition of ensuring normal communication functions [6]. However, the dimming control function may affect communication performance to some extent. Therefore, how to balance the dimming control function and the communication performance is a challenge for VLC systems. In order to meet this challenge, lots of researchers have proposed many dimming control schemes.

Variable on-off keying (VOOK) scheme and variable pulse position modulation (VPPM) scheme are two basic dimming control schemes proposed by the IEEE 802.15.7 task group [7]. The frames of the VOOK scheme can be divided into data frames and free frames. Data frames are utilized to realize communication, while free frames are responsible for dimming control. VPPM scheme is the combination of 2-pulse position modulation (2-PPM) and pulse width modulation (PWM), in which 2-PPM is used to realize data transmission, and PWM realizes dimming control. Paper [8] proposed multiple pulse position modulation (MPPM) scheme, which can further improve the spectral efficiency and error performance for dimmable indoor VLC systems. When *n* approaches infinity,

the MPPM scheme can reach the theoretical limit of transmission efficiency. Those modulation based schemes realize dimming control and data transmission through different modulation mode. A rate-compatible punctured code (RCPC) scheme has been proposed in [9]. The RCPC scheme can provide a wide range of brightness and simple coding structure for all different rates. Paper [10] provides a modified Reed-Muller (RM) code based scheme which utilizes a coset constructed from a bent function and RM codes for providing dimming control function. Paper [11] proposed a dimming control scheme based on weight threshold check code (WTCC), which can achieve dimming control and improve spectral efficiency further. The scheme is proposed for binary data transmission but can not provide constant transmission efficiency. The transmission efficiency will change when the dimming factor varies. Those coding-based schemes realize dimming control and data transmission by different encoding/decoding constructions. A dimming control scheme based on multilevel parity check codes (ML-PCC) proposed in paper [12], and paper [13] introduced a dimming control scheme based on multilevel incremental constant weight codes (ML-ICWC). Both of the two schemes are proposed for multilevel transmission and can not provide constant transmission efficiency. Significantly, the ML-PCC scheme can provide better error performance than the ML-ICWC scheme, but the dimming factor of ML-PCC can not cover the whole range.

However, all the dimming control schemes we mentioned above have the same problem: The transmission efficiency is not constant when the dimming factors are different. To solve this problem, paper [14] presents a block coding-based dimming control scheme that can provide constant transmission efficiency. Nevertheless, the dimming range and the error performance of the block coding-based scheme still has room for improvement. In this paper, we propose a dimming control scheme based on extensional constant weight codeword sets (ECWCS) to realize dimming control with constant transmission efficiency. The simulation results show that when the transmission efficiency and dimming factor are both the same, the error performance of the proposed scheme is better than that of the block coding scheme. We can also conclude that the proposed scheme has a wider dimming range than the block coding scheme. The ECWCS scheme can provide constant transmission efficiency, a wider dimming range, and a lower division value of the dimming factor. Therefore, the ECWCS scheme is a better choice for the indoor scene with variable brightness, such as office, conference room, classroom, and movie theater.

The rest of this paper can be settled as follows: Section 2 proposes the system model of the proposed ECWCS scheme. In Section 3, the motivation, codeword construction, and the encoding/decoding procedure of the proposed scheme are given. Section 4 shows the simulation results of the proposed scheme and the contrast scheme. The conclusion of the whole paper is shown in Section 6.

## 2. System Model

The system model of the proposed ECWCS scheme will be presented in this section. In this paper, the scheme is proposed for the single-input single-output VLC system, which contains a single LED and a single photodetector (PD). The system model is shown as Figure 1. First, the codeword set is constructed according to the dimming factor. After generated by the massage generator, the binary data is encoded by the dimming encoder. Then the coded binary data are modulated by the OOK modulator to the LED. After passing through the optical channel with additive white Gaussian noise (AWGN), the received signals are detected by the PD and then decoded by the dimming decoder. There is a point that should be noted: This scheme can not only be applied to OOK modulation but also pulse amplitude modulation (PAM) and multi-carrier modulation (MCM). Figure 1 is just an example of the ECWCS scheme applied to OOK modulation for the convenience of understanding. When the algorithm is applied to PAM, we should also adjust the average code weight of the codeword set to realize dimming control. Because PAM has a higher modulation order, the codeword set will be further expanded, and the spectral efficiency will be further increased. However, the corresponding error performance will

have a certain amount of loss. When the algorithm is applied to MCM, we should ensure the average code weight of the codeword set for every subcarrier is constant and adjust it to realize dimming control. When MCM is utilized in the proposed ECWCS scheme, the throughput will be increased, and the resistance to interference will be enhanced. However, the implementation complexity and cost will also be significantly increased. Modern Discrete Multi-Tone (DMT) modulation is a kind of MCM and can be utilized in the proposed scheme. Paper [15] reported the first visible light link based on WDM and DMT modulation of a single RGB-type white LED, and paper [16] proposed 1-Gb/s VLC systems using a white LED and the DMT signal. Both of the two schemes improved the throughput to a large extent. However, the implementation complexity and the cost of the electronic devices are weaknesses for the DMT modulation.

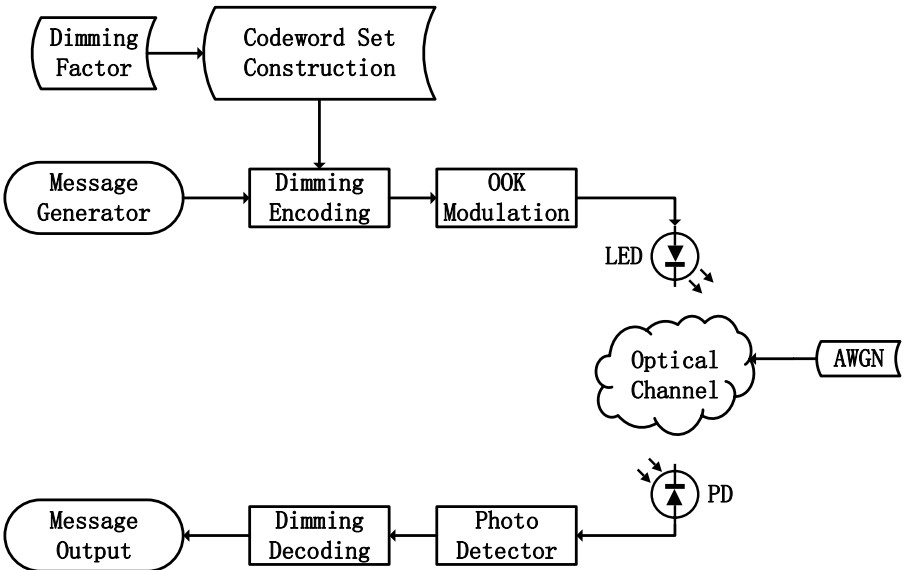

**Figure 1.** System model of the proposed dimming control scheme applied to on-off keying (OOK) modulation.

At last, the message will be output to the users. In this system model, there are a few assumptions that should be noted:

a. The channel state information (CSI) is available both at the receiver and the transmitter.
b. Compared with the direct light, the reflected light is much weaker in the indoor VLC systems [1,17]. For the convenience of computer analysis, we only consider the LOS path. Therefore, the multipath influence may not be considered in the proposed indoor dimmable VLC system.
c. A static link should be supposed in the proposed scheme because the channel is slow time-varying.

From Figure 1 and the assumptions we mentioned before, we can conclude that the received signal can be expressed as

$$y = \mu h x + n, \tag{1}$$

where $y$ represents the received signal, $x$ denotes the transmitted signal, $\mu$ is the photo-electric conversion factor which can be normalized as $\mu = 1$, $h$ is the channel gain we will introduce in this section, and $n$ is the additive white Gaussian noise of which the mean is 0 and the variance is $\sigma^2$.

From the contents in [1], the channel can be modeled as the Lambertian model, and the channel can be calculated by:

$$h = \begin{cases} \frac{A(u+1)}{2\pi D^2} \cos^u(\phi) T_S(\psi) g(\psi) \cos\psi, & 0 \leq \psi \leq \Psi_C \\ 0, & \psi > \Psi_C \end{cases} \tag{2}$$

where $h$ is the channel gain between the receiver and the transmitter, $D$ is the distance from the LED to the PD, $u$ is the order of the Lambertian model, and $A$ represents the physical area. The other physical quantities are about illumination. $\psi$ is the incidence angle, and $\phi$ denotes the irradiance angle. $T_S(\psi)$ represents the optical filter gain of the LED, $g(\psi)$ is the concentrator gain of the LED, and the field of view (FOV) of the PD can be represented by $\Psi_C$ [1].

For the dimmable VLC systems, there are two crucial constraints. The first constraint is the peak power constraint. The LEDs utilized for the VLC system have the optimal operating range due to the characteristics of nonlinear. We should also take the eye safety of the users into account [18]. Therefore, the peak power constraint is necessary, and it can be expressed as $0 \leqslant x \leqslant P_{peak}$, where $P_{peak}$ is the peak power. In this paper, we can normalize the peak power without loss of generality, which can be expressed as $P_{peak} = 1$. So the normalized power limitation is

$$0 \leqslant x \leqslant 1. \tag{3}$$

The other constraint is the dimming control constraint. In essence, the dimming control function of the peak power limited VLC systems is to adjust the average power when the peak power is fixed. The dimming factor is the ratio of the average power to peak power. We adjust the average power to meet the requirements of the users by communication technologies [9]. Therefore, the dimming control constraint can be expressed as

$$\bar{P} = E(\mathbf{x}) = \gamma P_{peak}, \tag{4}$$

where $\gamma$ denotes the dimming factor of the dimmable VLC systems with the range $\gamma \in (0, 1)$.

## 3. The Proposed Dimming Control Scheme

This paper introduces the motivation, the codeword set construction, and the encoding/decoding procedure of the proposed ECWCS scheme.

### 3.1. Motivation of the ECWCS Scheme

In order to realize dimming control for VLC systems with constant transmission efficiency, paper [14] proposed a scheme based on block coding via the bitwise AND operation. The error performance and spectral efficiency still have room for improvement. Motivated by the works in [12,13], we propose a dimming control scheme for VLC systems with constant transmission efficiency. In the proposed scheme, the code weight of the codewords in a set are all odd or all even. Therefore, the minimum Hamming distance of a codeword set is 2, which can improve the error performance to a certain extent. Meanwhile, the extensional constant weight codeword set scheme can achieve arbitrary dimming control within a wide range.

### 3.2. Construction of the Codeword Set

In this subsection, we provide the construction of the codeword set. Firstly, some notations should be defined. In the proposed ECWCS scheme, the transmitted binary data should be divided into several binary data sequences with the same length. We define the length of the original binary sequence is $k$, thus the original binary sequence can be expressed as $\mathbf{b} = [b_1, b_2, \cdots, b_k]$. The length of the dimming coded binary sequence is defined as $n$, thus the coded binary sequence is represented by $\mathbf{c} = [c_1, c_2, \cdots, c_n]$. From the definitions provided above, it is not difficult to know that the most important point of the

proposed scheme is to find the proper value of $n$ and construct the codeword set when $k$ is known.

We know that when $i$ is a positive integer and $i \leqslant n/2$, the combinatorial number have the relationship as follows: $C_n^1 < C_n^2 < \cdots < C_n^i$, where $C_n^m$ represents the combinatorial number of selecting $m$ elements from $n$ elements. The essence of the proposed scheme's codeword set construction is to increase the number of codewords available while keeping the transmission efficiency constant and dimming control function. In order to ensure the dimming control function, the average code weight of a codeword set should be fixed. In order to obtain a constant transmission efficiency, the ratio of $k$ to $n$ should be a constant when the dimming factor $\gamma$ varies. Based on the above two points, when the length of the original binary sequence $k$ is fixed, the length of the coded binary sequence need to satisfy the condition:

$$n \geqslant 2^k. \tag{5}$$

We define $\mathcal{S}$ is the codeword set constructed by the proposed scheme, and $m$ represents the total number of bit '1' of all elements in set $\mathcal{S}$. Thus the dimming factor can be expressed as

$$\gamma = \frac{m}{n \times 2^k}. \tag{6}$$

From Equation (6) and the analyses in this subsection, we know that the dimming range of the proposed dimming control scheme is $[\frac{1}{n}, \frac{n-1}{n}]$ and the division value of the dimming factor is $\frac{1}{n \times 2^{k-1}}$. The reason for the division value is $\frac{1}{n \times 2^{k-1}}$ rather than $\frac{1}{n \times 2^k}$ is as follows: In order to improve the error performance, we make all the code weight of the codewords in the codeword set odd or even to ensure the Hamming distance of the codewords in the codeword set is 2. For example, if 111000 is a codeword in the codeword set $\mathcal{S}$, the code weight of the rest codewords in $\mathcal{S}$ are all odd. Meanwhile, $2^k$ is an even, and $m$ is the total number of bit '1' of all elements in set $\mathcal{S}$. In other words, it represents the total code weight of codewords in $\mathcal{S}$. The summation of $2^k$ odd numbers (or even numbers) is an even number, thus the division value of the dimming factor is $\frac{2}{n \times 2^k} = \frac{1}{n \times 2^{k-1}}$. For the sake of calculation, in the rest of this paper, the dimming factor $\gamma = a \times \frac{1}{2^k}$, where $a$ is an even integer with $0 < a \leqslant 2^k \times \frac{n-1}{n}$. According to the value of the dimming factor, the encoding procedure can be divided into two cases:

For the convenience of expression, we define $i$ is a positive integer and $0 < i < n$.

(1) When the dimming factor $\gamma = \frac{i}{n}$, the code weight of every codeword in the codeword set $\mathcal{S}$ is $i$. Therefore, we can not only guarantee the dimming factor $\gamma = \frac{i}{n}$ but also ensure the Hamming distance of the codewords in the codeword set $\mathcal{S}$ is 2 to improve the error performance.

(2) When the dimming factor $\gamma \neq \frac{i}{n}$ and $\frac{1}{n} < \gamma < \frac{n-1}{n}$, the construction of the codeword set is more complicated. First we find the range $\frac{i}{n} < \gamma < \frac{i+1}{n}$. Then we encode the original binary codes like the last case which makes all the code weight of every codeword in set $\mathcal{S}$ is $i$. At last, we calculate $m = n \times 2^k \times \gamma$, $q = (m - 2^k i)/2$, select $q$ codewords in $\mathcal{S}$ and replace two bit '0' with two bit '1' of the $q$ codewords respectively.

To make it easier to understand, we fix $k = 3$ provide two examples for the two cases. When $k = 3$, we can calculate $n \geqslant 8$ from Equation (5), thus we fix $n = 8$. The examples are as follows:

**Example 1.** *When the dimming factor $\gamma = 1/8$, $i = 1$. Therefore, the mapping between the original binary sequence and the dimming coded binary sequence is shown in Table 1.*

**Table 1.** The mapping between the original binary sequence and the dimming coded binary sequence with $k = 3$, $n = 8$, and $\gamma = 1/8$.

| Original Binary Sequence | Dimming Coded Binary Sequence | Original Binary Sequence | Dimming Coded Binary Sequence |
|---|---|---|---|
| 000 | 00000001 | 100 | 00010000 |
| 001 | 00000010 | 101 | 00100000 |
| 010 | 00000100 | 110 | 01000000 |
| 011 | 00001000 | 111 | 10000000 |

**Example 2.** *When the dimming factor $\gamma = 5/16$, we firstly find the range $2/8 < 5/16 < 3/8$ and conclude that $i = 2$. Then we construct a codeword set $\mathcal{S}$ with $i = 2$ like the last example and calculate that $m = n \times 2^k \times \gamma = 8 \times 2^3 \times 5/16 = 20$, $q = (m - 2^k i)/2 = (20 - 2^3 \times 2)/2 = 2$. Thus we select 2 codewords in $\mathcal{S}$ and replace two bit '0' with two bit '1' of the 2 codewords respectively. Therefore, the mapping between the original binary sequence and the dimming coded binary sequence is shown in Table 2.*

**Table 2.** The mapping between the original binary sequence and the dimming coded binary sequence with $k = 3$, $n = 8$, and $\gamma = 5/16$.

| Original Binary Sequence | Dimming Coded Binary Sequence | Original Binary Sequence | Dimming Coded Binary Sequence |
|---|---|---|---|
| 000 | 00000011 | 100 | 00110000 |
| 001 | 00000110 | 101 | 01100000 |
| 010 | 00001100 | 110 | 11000011 |
| 011 | 00011000 | 111 | 10000111 |

The construction of the codeword set can be summarized in Algorithm 1.

---

**Algorithm 1:** Construction of the Codeword Set.

**Input**:The dimming factor $\gamma$, the length of original binary sequence $k$ and the original binary signal **s**

Fix proper the length of the coded binary sequence $n$ by the constraint: $n \geqslant 2^k$.

**If** $\gamma = \frac{i}{n}$, where $i$ is a positive integer and $0 < i < n$

Construct the codeword set $\mathcal{S}$ in which the code weight of every codeword is $i$.

**Else If** $\gamma \neq \frac{i}{n}$ and $\frac{1}{n} < \gamma < \frac{n-1}{n}$

Find the range $\frac{i}{n} < \gamma < \frac{i+1}{n}$

Construct the codeword set $\mathcal{S}$ in which the code weight of every codeword is $i$.

Calculate $m = n \times 2^k \times \gamma$ and $q = (m - 2^k i)/2$

Select $q$ codewords in $\mathcal{S}$ and replace two bit '0' with two bit '1' of the $q$ codewords respectively.

**End**

**End**

**Output**: The codeword set $\mathcal{S}$

---

### 3.3. Encoding/Decoding Procedure the ECWCS Scheme

In the last subsection, the codeword set $\mathcal{S}$ is constructed by Algorithm 1. By the construction of the codeword set, we can obtain the mapping relation between the original binary sequence and the dimming coded binary sequence from the table like Tables 1 and 2. In this subsection, the encoding/decoding procedure will be introduced.

For the encoding procedure, we first divide the original binary signal **s** into several length-$k$ binary sequences. Then we map the binary sequence **b** to the length-$n$ dimming coded binary sequence **c**. In the end, we can obtain the transmitted binary signal $x$.

For the decoding procedure, paper [19] provide a fast decoding algorithm. Motivated by the contents in paper [19], we provide a decoding algorithm to decrease complexity. We know that the traditional Maximum Likelihood (ML) decoding algorithm is to compare

the probability density function of the received sequence **r** conditioned on the coded binary sequence **c** when the system is a single-input single-output system, which is expressed as [20]:

$$p(\mathbf{r}|\mathbf{c}) = \frac{1}{(\sqrt{2\pi\sigma^2})^N}\exp(-\frac{\|\mathbf{r}-\mathbf{c}\|^2}{2\sigma^2}). \tag{7}$$

The decoder can be simplified as:

$$\hat{\mathbf{r}} = \arg\min\|\mathbf{r}-\mathbf{c}\|^2. \tag{8}$$

In order to decrease the complexity of the decoding procedure, the decoder can also be given as:

$$\|\mathbf{r}-\mathbf{c}\|^2 = \|\mathbf{r}\|^2 + \|\mathbf{c}\|^2 - 2\mathbf{r}^T\mathbf{c}. \tag{9}$$

When the received signal is the same one and the code weight is a constant, the decoder is represented by:

$$\hat{\mathbf{r}} = \arg\max\mathbf{r}^T\mathbf{c}. \tag{10}$$

Due to the coded sequence **c** is a binary sequence, we should only find the position of bit '1' in **c** and sum the elements at the same position in **r**. For example, when $\mathbf{c} = [1,1,0,0]$ and $\mathbf{r} = [1.05, 0.83, 0.35, 0.21]$, the summation is $1.05 + 0.83 = 1.88$. After we get the estimated sequence $\hat{\mathbf{r}}$ of **r**, we can recover the original binary data by looking up from the table like Tables 1 and 2.

The proposed decoding algorithm is for the constant weight codes. Therefore, according to the two cases in the last subsection, there are two cases for the decoding algorithm. The two cases are as follows:

(1)    When the dimming factor $\gamma = \frac{i}{n}$, the code weight of every codeword in the codeword set $\mathcal{S}$ is $i$. Thus we can utilize the proposed decoding algorithm directly.

(2)    When the dimming factor $\gamma \neq \frac{i}{n}$ and $\frac{1}{n} < \gamma < \frac{n-1}{n}$, the code weight of every codeword in set $\mathcal{S}$ is not a constant. First we find the range $\frac{i}{n} < \gamma < \frac{i+1}{n}$ and calculate $m = n \times 2^k \times \gamma, q = (m - 2^k i)/2$. Then we find the $q$ codewords the code weight of which is not $i$ in $\mathcal{S}$. At last, replace two bit '1' with two bit '0' of the $q$ codewords respectively. That is the inverse process of the codeword set construction and requires the decoder to know the details of the construction process. In this way, we can utilize the proposed decoding algorithm.

The decoding algorithm can be summarized in Alogrithm 2.

---

**Algorithm 2:** Decoding Procedure.

**Input:** The dimming factor $\gamma$, the length of original binary sequence $k$,
    the length of received binary sequence $n$, and the received sequence **r**
**If** $\gamma \neq \frac{i}{n}$ and $\frac{1}{n} < \gamma < \frac{n-1}{n}$
  Find the range $\frac{i}{n} < \gamma < \frac{i+1}{n}$
  Calculate $m = n \times 2^k \times \gamma$ and calculate $q = (m - 2^k i)/2$
  Replace two bit '1' with two bit '0' of the $q$ codewords respectively.
**Else If** $\gamma = \frac{i}{n}$, where $i$ is a positive integer and $0 < i < n$
    **End**
**End**
Find the position of bit '1' in **c**, sum the elements at the same position in **r**, and obtain the estimated sequence $\hat{\mathbf{r}}$ of **r**.
Recover the original binary data by looking up from the table like Tables 1 and 2.
**Output:** The estimated signal $\hat{\mathbf{s}}$ of the original binary signal **s**.

---

## 4. Simulation Results

In this subsection, we will provide the simulation results, including dimming range, error performance, and spectral efficiency. The analysis will be provided at last.

### 4.1. Dimming Range

Dimming range is the range that can be achieved by a dimming control scheme, which can be expressed as:

$$\gamma_r = \gamma_{max} - \gamma_{min},\tag{11}$$

where $\gamma_{min}$ is the minimum dimming factor, $\gamma_{max}$ represents the maximum dimming factor, and $\gamma_r$ is the dimming range.

For both the two dimming control schemes with constant transmission efficiency, including the block coding scheme and the proposed ECWCS scheme, the dimming range depends on the value of $n$ and the value of $n$ is affected by the value of $k$. Thus we provide the relationship between $k$ and the dimming range $\gamma_r$. What needs illustration is that the length of the coded binary sequence $n$ of the proposed ECWCS scheme is not unique when $k$ is fixed. For convenience, we utilize $n = 2^k$. In paper [14], the dimming range of the block coding dimming control scheme can be summarized as Table 3, and the dimming factor's division value of the block coding scheme is $\frac{1}{k}$.

**Table 3.** The dimming range of the block coding scheme under different $k$.

| $k$ | $\gamma_{min}$ | $\gamma_{max}$ | $\gamma_r$ |
|---|---|---|---|
| 2 | $\frac{1}{4}$ | $\frac{3}{4}$ | $\frac{1}{2}$ |
| 3 | $\frac{1}{6}$ | $\frac{5}{6}$ | $\frac{2}{3}$ |
| 4 | $\frac{1}{8}$ | $\frac{7}{8}$ | $\frac{3}{4}$ |
| $k$ | $\frac{1}{2k}$ | $\frac{2k-1}{2k}$ | $\frac{k-1}{k}$ |

The dimming range of the proposed ECWCS dimming control scheme is shown in Table 4.

**Table 4.** The dimming range of the proposed ECWCS scheme under different $k$.

| $k$ | $\gamma_{min}$ | $\gamma_{max}$ | $\gamma_r$ |
|---|---|---|---|
| 2 | $\frac{1}{4}$ | $\frac{3}{4}$ | $\frac{1}{2}$ |
| 3 | $\frac{1}{8}$ | $\frac{7}{8}$ | $\frac{3}{4}$ |
| 4 | $\frac{1}{16}$ | $\frac{15}{16}$ | $\frac{7}{8}$ |
| $k$ | $\frac{1}{2^k}$ | $\frac{2^k-1}{2^k}$ | $\frac{2^{k-1}-1}{2^{k-1}}$ |

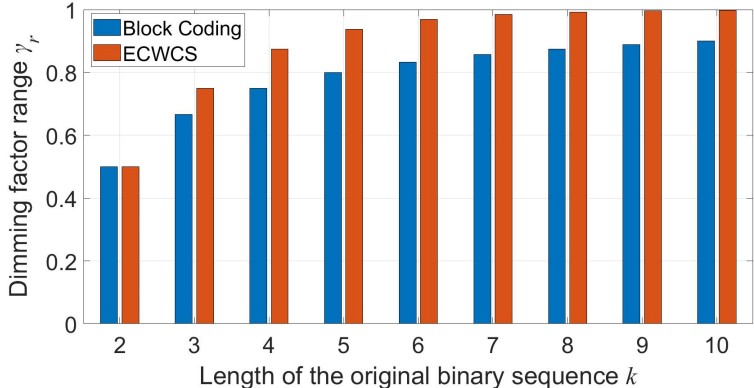

**Figure 2.** Dimming range of the block coding scheme and the extensional constant weight codeword sets (ECWCS) scheme under different $k$.

Figure 2 shows the dimming range under different $k$. From Tables 3 and 4 and Figure 2, we can know that the proposed ECWCS scheme has a wider dimming range and a lower division value compared with the block coding scheme. We can also see that under the condition of $k \geqslant 7$, the dimming range is nearly all covered when the ECWCS scheme is utilized.

*4.2. Error Performance*

Error performance is another essential index for VLC systems. It is represented by curves of the bit error rate (BER). The curves describe the relationship between the signal-to-noise ratio (SNR) and BER. The SNR can be expressed as [21]:

$$\text{SNR} = 10 \lg \frac{1}{R_C \sigma^2},\tag{12}$$

where $R_C$ is the code rate.

In this paper, we compare the block coding dimming control scheme and the proposed scheme. From the discussion before and the contents in the last subsection, we know that the length of the coded binary sequence $n$ of the proposed ECWCS scheme is not unique when $k$ is fixed. As shown in Equation (5), for the proposed scheme $n \geqslant 2^k$. While for the block coding scheme, $n = k^2$. Therefore, we fix the value of $n$, $k$, and $\gamma$ to compare the error performance of the two schemes for fairness.

From Figure 3, we can conclude that when the value of $n$, $k$, and $\gamma$ are the same, the error performance of the ECWCS scheme is better than that of the block coding scheme. That is because the minimum Hamming distance between the codewords of the ECWCS scheme is larger than that of the block coding scheme. We can also see that for the proposed ECWCS scheme, the error performance with $k = 2, n = 4$ outperformed $k = 3, n = 9$ under the same dimming factor.

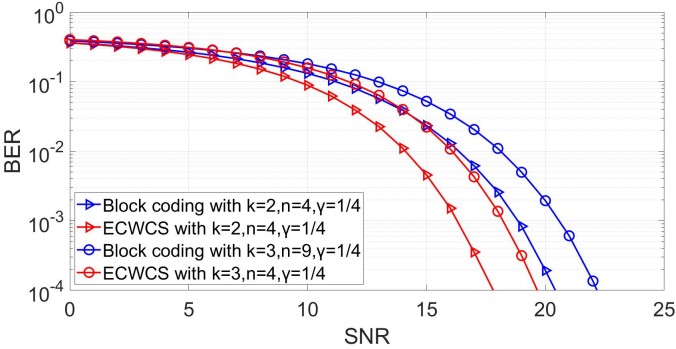

**Figure 3.** Error performance of the block coding scheme and the ECWCS scheme.

For clarity, we demonstrate the adaptability of this system in terms of provided illuminance. First, we introduce the relationship between the dimming factor and illuminance. We know that the dimming factor is the ratio of the average power to peak power, and the range of dimming factor is $0 < \gamma < 1$. The value of the dimming factor reflects the illuminance of the LED. With the increase of the dimming factor, the illuminance of the LED increases gradually. Then we demonstrate the relationship between BER and illuminance. We can assume a set $\mathcal{D}$ consists of the Euclidean distance between any two distinct received signals. The minimum Euclidean distance (MED) of the received signals can be defined as the minimum of the elements in set $\mathcal{D}$. Therefore, the MED of the received signals can be expressed as:

$$d_R = \min \sqrt{\|y^a - y^b\|^2},\tag{13}$$

where $y^a$ and $y^b$ are two distinct received signals.

Similarly, the MED of the transmitted signals is:

$$d_T = \min \sqrt{\|x^a - x^b\|^2},\tag{14}$$

where $x^a$ and $x^b$ are two distinct transmitted signals.

We know that the received signal can be expressed as Equation (1) in Section 2. We substitute Equation (1) into Equation (13) and the MED of the received signals can be also expressed as

$$d_R = \min \sqrt{\|(\mu h x^a + n) - (\mu h x^b + n)\|^2} = \min \sqrt{\mu h \|x^a - x^b\|^2} = d_T \sqrt{\mu h}.\tag{15}$$

There is a point that should be noted: In terms of sequences, the received signal is the received sequence **r** and the transmitted signal is the coded binary sequence **c**. The distance between the signals means the distance between the sequences, and the numbers in the sequence represent the intensity level of the optical signal. From the contents in paper [22], we know that the BER curve can represent the error performance, and the error performance is determined by the MED of the received signals when the SNR is fixed. Therefore, in the proposed system model of this paper, the error performance is determined by the MED of the transmitted signals when the SNR is fixed. The way we explain the BER metric according to minimum Euclidean distance is not only for OOK modulation but also for pulse amplitude modulation. In this paper, we provide OOK modulation as an example to introduce the ECWCS scheme. For the proposed ECWCS scheme, the MED of the transmitted signals is constant when the dimming factor varies. Therefore, the BER curves are the same with different dimming factors. However, when the dimming factor varies, the average power of the transmitted signals changes. The SNR will be affected by the change of the average power of the transmitted signals. With the increase of the average power, SNR increases gradually. Therefore, the error performance will be affected by the dimming factor. From the above reasoning and the BER curves provided in Figure 3, we can conclude that with the increase of the dimming factor, the error performance gets better gradually.

### 4.3. Spectral Efficiency

The dimming control scheme based on extensional constant weight codeword sets is proposed for indoor VLC systems with constant transmission efficiency in this paper. The other dimming control schemes widely utilized can not realize constant transmission efficiency when the dimming factor varies. Spectral efficiency indicates the effective bit rate $R_b$ can be realized when the bandwidth $B$ is fixed and can be expressed as:

$$\nu = \frac{R_b}{B}.\tag{16}$$

When the bandwidth $B$ is fixed, the spectral efficiency is only related to the effective bit rate $R_b$. Therefore, the spectral efficiency can reflect the transmission efficiency and the effective throughput. The proposed ECWCS scheme guarantee that the spectral efficiency is not affected by the variation of the dimming factor through the unique way of encoding. Therefore, we can say that the proposed ECWCS scheme realizes constant spectral efficiency.

Figure 4 shows the spectral efficiency comparison between the proposed ECWCS scheme and other dimming control schemes without constant transmission efficiency when the length of the coded binary sequence $n = 8$.

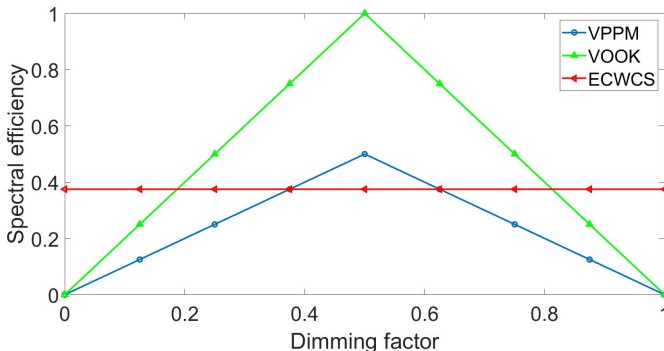

**Figure 4.** Spectral efficiency of variable on-off keying (VOOK), variable pulse position modulation (VPPM) and ECWCS.

We can see from Figure 4 that the spectral efficiency of the VOOK scheme and the VPPM scheme is greater than that of the ECWCS scheme within a specific dimming range. However, when the dimming factor is close to 0 or 1, the spectral efficiency of the ECWCS scheme is greater than that of the VOOK scheme and the VPPM scheme. It can be construed that the constant transmission efficiency is guaranteed by reducing the spectral efficiency within a specific dimming range.

We can assume a scenario that the users need to change the dimming factor of the indoor VLC system from 0.1 to 0.5. When we utilize the VOOK scheme or the VPPM scheme, the transmission efficiency varies according to the value of the dimming factor, and the communication quality would be influenced. On the other hand, for example, when the length of the coded binary sequence $n = 8$, the dimming factor's division value of the VOOK scheme and the VPPM scheme is $1/8$. From the contents in Section 3, we know that the dimming factor's division value of the proposed ECWCS scheme is $\frac{1}{n \times 2^{k-1}} = 1/32$. Therefore, based on the above two points, the proposed ECWCS scheme is a better choice for the indoor scene with variable brightness.

*4.4. Analysis*

In this section, we have provided a comparison to the currently existing variable-weight coding scheme, which is called block coding dimming control scheme [14]. The code weight of the block coding dimming control scheme is variable when the dimming factor is fixed. Thus we can classify it as the variable-weight coding scheme. We can know that compared with the block coding dimming control scheme, the ECWCS scheme has a wider dimming range, a smaller division value, and better error performance.

The currently existing constant-weight schemes can not provide a constant transmission. All of those schemes have the same code weight when the dimming factor is fixed. However, when the dimming factor varies, the code weight and transmission efficiency will change. Thus it is unfair to compare those schemes with the proposed ECWCS scheme. When we utilize constant-weight codes to realize dimming control with constant transmission efficiency, it will add redundancy compared with the ECWCS scheme. For example, we fix the dimming factor is $\gamma = 3/8$ and the length of the original binary sequence is $k = 2$. The length of the coded binary sequence is $n = 4$ when the ECWCS scheme is utilized, and the length of the coded binary sequence is $n = 8$ when utilizing constant-weight codes.

## 5. Discussion

We have done much research on the dimming control function of VLC systems in different scenes. Paper [12] proposed a dimming control scheme based on multilevel parity check codes (ML-PCC) for multilevel transmission. A dimming control scheme based on multi-LED phase-shifted space-time codes (MP-STC) were proposed for multi-LED VLC systems in paper [23]. Paper [24] provided a dimming control scheme based on constant weight space-time codes (CWSTC) for Multiple Input Multiple Output (MIMO) VLC systems. In this paper, we propose the ECWCS scheme for the dimmable VLC systems with

constant transmission efficiency and a low complexity decoding algorithm for the ECWCS scheme. In addition to this, it can also provide a wider dimming range, a smaller division value, and better error performance compared with the previous constant transmission efficiency dimming control scheme [14]. We think that the ECWCS scheme is a better choice for dimmable VLC systems with constant transmission efficiency.

## 6. Conclusions

In this paper, a dimming control scheme based on extensional constant weight codeword sets has been proposed for the constant transmission efficiency VLC systems. The proposed ECWCS scheme can realize reliable transmission and maintain constant transmission efficiency when the dimming factor varies by optimizing the codeword set. From the simulation results, the proposed ECWCS scheme has a wider dimming range, a smaller division value, and better error performance compared with the block coding dimming control scheme. Meanwhile, a low complexity decoding algorithm has been proposed for the ECWCS scheme to enhance practicality. Therefore, the ECWCS scheme will be considered as a potential choice for the constant transmission efficiency dimmable VLC systems.

**Author Contributions:** J.-N.G. has contributed to the scientific part of this work. J.-N.G. and J.Z. contributed to the conception and design of the study. G.X. and L.L. have critically reviewed the paper. All authors have read and agreed to the published version of the manuscript.

**Funding:** This work is supported by Grant 161100210200 from Major Scientific and Technological of Henan Province, China, Grant 2016B010111001 Major Scientific and Technological of Guangdong Province, China, the National Key Research and Development Project (2018YFB1801903) and the National Natural Science Foundation of China (NSFC) under Grant (62071489,61671477,61901524).

**Acknowledgments:** The authors wish to thank the anonymous reviewers for their valuable suggestions.

**Conflicts of Interest:** The authors declare no conflict of interest.

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
