# Peer review of "Constant Transmission Efficiency Dimming Control Scheme for VLC Systems"

_photonics, doi:10.3390/photonics8010007_

Round 1

Reviewer 1 Report

The authors propose a novel dimming scheme to keep the transmission efficiency constant. The paper is well written and the authors express the novel ideas quite clear. However, there are few comments that I would like authors to address before its publication:

1) The simulation results of the paper are limited. It seems to be focused to a conference paper rather than a journal paper. I suggest to get more extensive simulation results. E.g.:
How are other factors such as complexity or rate against counterparts that do not necessarily keep transmission efficiency constant? Maybe this algorithm focus on keep transmission efficiency but forget about the rest. Thus, I require to do a comparison with state-of-the-art techniques although they do not guarantee a constant efficiency.

2) This algorithm can only be applied to OOK? I would like authors explain this as it may be a limitation. Otherwise, it is required an explanation of how this technique could be adapted to be used with other modulation schemes.

3) I would like that authors demonstrate the adaptability of this system in terms of provided illuminance. How BER and illuminance is affected for different dimming factors? It would be good to get these results in a realistic scenario.

4) There are few typos, such as after eq. (2) "\phi denotes the incidence angle" instead of irradiance angle. The paper deserves another proofreading.

Reviewer 2 Report

  1. The proposed encoding scheme improves performance in terms of bit error, but adds redundancy. It remains to explain, or perform simulations, etc., indicating how they affect the effective throughput of the VLC link.
  2. Missing citations or references for equations (2), (7) and (12)

Reviewer 3 Report

General notes:

Please clearly state the differences between the current work and your previous works in Refs. [11,15,16] and consider providing performance comparison.

Please provide a reference/comparison to the currently existing constant-weight/variable-weight coding schemes.

line 71: LOS path presence or the only LOS path? What about multipath that can significantly influence high-speed OOK performance? What about modern DMT modulations that provide significantly higher throughput?

typo in line 120: What is the relation about?

Round 2

Reviewer 1 Report

The authors have considerably improved the quality of the paper, but still there are some points that deserve clarification:

1) The following sentence is not very accurate: "We know that the BER curve can represent the error performance, and the error performance is determined by the minimum Euclidean distance (MED) of the received signals [22]". Euclidean distance between what?

2) Also, this sentence "Therefore, the BER curve is not affected by different dimming factors" does not seem to be correct. Changing the dimming factor means to change the average output luminous flux of the LED, which for sure affects the SNR and as a consequence the BER on the receiver side. Something seems to be wrong in authors' reasoning.

Reviewer 3 Report

In the expanded Section 4.2 and newly added Section 4.3, there is something that seems like a contradiction:

  • constant spectral efficiency
  • constant bandwidth
  • variable dimming, i.e. change in illuminance of the LED = change in the transmitted power
  • "BER curve is not affected by different dimming factors"

Something more in the above has to be variable as well, otherwise, these statements contradict each other.

Round 3

Reviewer 1 Report

The authors addressed the comments of this reviewer in a coherent way.

Please, consider these further comments for the review:

1) The way authors explain the BER metric according to minimum Euclidean
distance is very generic. To this reviewer's opinion this is only valid
for OOK modulation schemes and authors should indicate this, as well as
the fact that, for distance, they mean the high and low signal levels.

2) These two sentences contradict and the authors should correct them
according to results:
"Therefore, the BER curves are the same with different dimming factors."
and "Therefore, the error performance will be affected by the
dimming factor"

For the rest, the authors addressed the comments reviewer properly.

Reviewer 3 Report

The added explanations clarify the point arisen by the reviewer. The recent clarification by authors showed that the impact of the paper is even lower than I have expected. The variable BER coding is mostly unuseful. 

Author Response

This manuscript is a resubmission of an earlier submission. The following is a list of the peer review reports and author responses from that submission.